# Changes in Faecal and Plasma Amino Acid Profile in Dogs with Food-Responsive Enteropathy as Indicators of Gut Homeostasis Disruption: A Pilot Study

**DOI:** 10.3390/vetsci10020112

**Published:** 2023-02-03

**Authors:** Cristina Higueras, Rosa Escudero, Almudena Rebolé, Mercedes García-Sancho, Fernando Rodríguez-Franco, Ángel Sainz, Ana I. Rey

**Affiliations:** 1Departamento de Producción Animal, Facultad de Veterinaria, Universidad Complutense de Madrid, Avda. Puerta de Hierro s/n., 28040 Madrid, Spain; 2Departamento de Medicina y Cirugía Animal, Facultad de Veterinaria, Universidad Complutense de Madrid, Avda. Puerta de Hierro s/n., 28040 Madrid, Spain

**Keywords:** canine, chronic enteropathy, food-responsive enteropathy, faecal amino acids, plasma amino acids

## Abstract

**Simple Summary:**

Food-responsive enteropathy (FRE) has the greatest prevalence among the different groups of chronic enteropathies. However, information is lacking on the specific amino acid profile for FRE in dogs and its diagnostic utility. This study evaluated differences in the plasma and faecal amino acid profile between control and FRE in dogs as possible indicators of disease. We also searched for correlations between amino acids and parameter indicators of gut health, including body condition score (BCS), and indices, such as canine inflammatory bowel disease activity index (CIBDAI), to evaluate whether the amino acid profile could serve as an indicator of disease severity. Several alterations were observed in plasma and faecal amino acid profiles in sick dogs, and high correlations were found between amino acids and disease activity index or faecal characteristics. More information on the amino acid profile in dogs with FRE could help with diagnoses and lead to more precise and specific amino acid formulation, dietary interventions, better response to diet, and recovery of animals.

**Abstract:**

Dogs suffering from food-responsive enteropathy (FRE) respond to an elimination diet based on hydrolysed protein or novel protein; however, studies regarding the amino acid profile in FRE dogs are lacking. The aim of this pilot study was to evaluate whether the plasma and faecal amino acid profiles differed between control and FRE dogs and whether these could serve as indicators of severity of illness. Blood, faecal samples, body condition score, and severity of clinical signs based on the canine inflammatory bowel disease activity index were collected before starting the elimination diet. FRE dogs had lower proportions of plasma Asparagine, Histidine, Glycine, Cystine, Leucine, and branched-chain/aromatic amino acids; however, Phenylalanine increased. In faecal samples, Cystine was greater whereas Phenylalanine was lesser in sick dogs compared to control. Leucine correlated negatively with faecal humidity (r = −0.66), and Leucine and Phenylalanine with faecal fat (r = −0.57 and r = −0.62, respectively). Faecal Phenylalanine (r = 0.80), Isoleucine (r = 0.75), and Leucine (r = 0.92) also correlated positively with total short-chain fatty acids, whereas a negative correlation was found with Glycine (r = −0.85) and Cystine (r = −0.61). This study demonstrates the importance of Leucine and Phenylalanine amino acids as indicators of the disease severity in FRE dogs.

## 1. Introduction

Amino acids are important compounds in the organism as they constitute the main components of proteins and various bioactive molecules [1,2]. Recent studies have proved that amino acids play an important role in the gut and regulation of inflammation [3]. They participate in the proliferation and apoptosis of intestinal epithelial cells (IECs), expression of tight junction proteins (TJPs), inflammatory processes, and oxidative stress through the regulation of signalling pathways [4]. Consequently, amino acids can regulate this common process taking place in inflammatory diseases such as chronic enteropathies (CEs). Some studies carried out in human medicine have shown alterations of the amino acid profile in patients suffering from gastrointestinal diseases, such as inflammatory bowel disease (IBD), with decreased levels of several amino acids in serum samples and increased levels in faecal samples of these patients [5]. This finding also means that some amino acids could be used to monitor clinical disease activity [6] or even serve as part of the treatment by improving clinical symptoms [7].

In veterinary medicine, studies regarding metabolomics in CEs are focused on the search for potential biomarkers. It has been observed that cobalamin, folate, C-reactive protein or dysbiosis index (DI) could help in diagnostic evaluation, prognosis, and monitoring clinical activity [8]. However, studies regarding the nutritional profile of food-responsive enteropathy (FRE) in dogs are lacking. In previous research, we evaluated the short-chain and total fatty acid profiles in FRE dogs, finding alterations and correlations between some of these compounds and the intestinal disease activity index [9]. Currently, scientific reports concerning the amino acid profile in dogs with CEs are very limited, although a few studies have been conducted in dogs with immunosuppressant-responsive enteropathy (IRE) and protein-losing enteropathy (PLE), finding alterations in the amino acid profile when compared to healthy individuals [10,11,12,13]. Since FRE seems to have the greatest prevalence (60–70% of cases of CE) [14] and dogs suffering from this disease respond to an elimination diet based on hydrolysed protein or novel protein, the amino acid profile evaluation deserves more attention.

Thus, the aim of this study was, first, to compare the plasma and faecal amino acid profiles between control and FRE dogs in order to expand our current knowledge and characterize the disease. Our second aim was to search for correlations between amino acids and metabolites such as short-chain fatty acids (SCFAs) or other parameter indicators of gut health, including body condition score (BCS) or indices such as canine inflammatory bowel disease activity index (CIBDAI), to evaluate whether the amino acid profile could serve as an indicator of disease severity.

## 2. Materials and Methods

### 2.1. Animals and Sample Collection

All procedures and protocols were approved by the Animal Research Committee of the Veterinary Medicine Teaching Hospital, Complutense University of Madrid (reference number 11/2021). The owners of all the patients accepted their participation in the study through informed consent.

The criteria inclusion for healthy dogs (*n* = 6) were a normal physical examination, blood test, and the absence of any clinical signs, including digestive signs, for at least four months. Asymptomatic dogs with chronic diseases were excluded from the study. Only sick dogs (*n* = 9) that had been suffering from digestive clinical signs (weight loss, anorexia, hiporexia, vomiting, or diarrhoea) for at least three weeks were to be enrolled in the study. Moreover, they had to respond to an elimination diet based on novel protein or hydrolysed protein after one month of administration. Based on the successful response to the diet, they were diagnosed as dogs with FRE. No dog included in the study suffered from hypoproteinemia.

Data including sex, age, breed, sexual status, body weight, and BCS were collected from every patient (Table 1). Breeds of dogs with FRE were three mongrel dogs, one of each breed of Labrador Retriever, Cocker Spaniel, Miniature Schnauzer, Maltese, short-haired Dachshund, and Chihuahua. Breeds of healthy dogs included five mongrel dogs and one Gordon Setter. Information about specific digestive clinical signs was measured based on the CIBDAI, as previously described [15].

Blood samples (2 mL) were obtained following the regular procedure by jugular or cephalic venipuncture and collected in heparine tubes in fasted dogs for a period of at least 8 h. Plasma was obtained after centrifugation and stored at −80 °C. Faecal samples were collected by the owners after defecation (the same morning under fasting conditions) and brought to the clinic in less than three hours where they were stored at −20 °C. Blood and faecal samples of dogs with FRE were collected before starting the elimination diet. The commercial diets consumed by the dogs before the dietary treatment consisted of cereals, animal proteins, and vegetable/animal fats (averaged % according to manufacturer’s composition: humidity, 9.5 ± 0.0; crude protein, 26.8 ± 3.4; crude fat, 11.7 ± 4.4; ash, 5.9 ± 1.7; crude fibre, 1.8 ± 0.5; soluble fibre, 6.0 ± 0.7; nitrogen-free extractives, 38.3 ± 12.5; Ca, 0.9 ± 0.1; P, 0.7 ± 0.1; C18:2, 2.9 ± 1.1; ∑n-6, 2.7 ± 1.2; ∑n-3, 0.7 ± 0.1; mg/kg vitamin E: 619.2 ± 245.5; metabolic energy/1000 g: 3332.3 ± 645.2).

### 2.2. Concentration of Free Amino Acids in Plasma Samples

Plasma-free amino acids were extracted as described elsewhere [16]. Essentially, plasma samples (100 µL) were mixed with 500 µL of a mixture of acetronile:methanol:acetone. After centrifugation, the supernatant was removed, leaving the dry residue. The supernatant was evaporated in N_2_ stream, redissolved in 500 µL water (MilliQ) and stored at −20 °C until analysis. The plasma amino acids and their standards were then derivatised with OPA (*o*-phtalaldehyde) as described by Jones et al. [17]. Samples were derivatised in an HPLC (Hewlet-Packard 1100 Agilent Technologies GmbH, Walbronn, Germany) equipped with a fluorescent detector, a phase reverse column Porshell HPH-C18 (4.6 × 100 mm, 2.7 µm, Agilent Technologies, Walbronn, Germany), and a pre-column HPH-C18 (Infinitylab Porshell 120, 3.0 mm, UHPLC, Agilent Technologies, Germany). Two mobile phases were used: phase A, a dilution of 10 mM Na_2_HPO_4_, 10 mM Na_2_B_4_O_7_ pH 8.2, and 0.5 mM NaN_3_; and phase B, a mixture of acetonitrile:methanol:water. The detector was adjusted at 340 nm for excitation and 450 nm for emission. The determination of amino acids was made by comparing their retention times with those of a standard sample of nineteen amino acids: aspartic acid (Asp), glutamic acid (Glu), serine (Ser), alanine (Ala), arginine (Arg), cystine (Cys-Cys), histidine (His), glycine (Gly), leucine (Leu), isoleucine (Ile), lysine (Lys), methionine (Met), threonine (Thr), phenylalanine (Phe), tyrosine (Tyr), and valine (Val) (1 nm/µL in 0.1 M HCl, Agilent Technologies), along with a dilution of asparagine (Asn), glutamine (Gln) (0.01 M HCl), and tryptophan (Trp) (0.1 M HCl, Agilent Technologies).

### 2.3. Concentration of Amino Acids in Faecal Samples by Acid Hydrolysis

Lyophilized samples (50–80 mg) (Lyoquest, Telstar, Tarrasa, Spain) were placed in screw-capped glass tubes and hydrolysed with 15 mL of 6 M HCl. These tubes were then flushed with N_2_ and heated to 110 °C for 22 h. After cooling at room temperature, samples were filtered through filter paper to a beaker, and the pH was adjusted to 5.6 by the addition of NaOH solution (phmeter Crison Basic 20+). The solution was placed in a 100 mL volumetric flask and levelled up to that volume. Then, 20 mL were collected with a syringe and filtered by Sep-pak silica cartridge. Subsequently, 2 mL of the sample extract was isolated in a vial and stored at −20 °C. Protein hydrolysates and an amino acid calibration mixture were derivatised by o-phtaldialdehyde. Finally, an analysis of these samples was properly carried out by HPLC under the same conditions previously described by plasma samples analysis.

### 2.4. Statistical Analysis

For the analysis of variance, data were analysed following a completely randomised design using the general linear model (GLM) procedure contained in SAS (version 9; SAS Inst. Inc., Cary, NC, USA) following the model: Y_ij_ = µ + T_i_ + ξ_ij_ (where Y is the data observed of the dog j of the status i, µ is the average, T is the dog status (i = 1, 2), and ξ is the residual error). Data were presented as the mean of each group and the standard deviation of the mean (SD) together with significance levels (*p* values). Differences were considered significant at *p* < 0.05. Pearson correlation among different amino acids and condition indices or other components of plasma/faeces, such as humidity, fat, and α-tocopherol (determined in a previous paper [9]), were carried out using the Statgraphics-19 program. The linear adjustments between amino acids and faecal characteristics or SCFAs (analysed in a previous study) [9] were quantified by Statgraphics-19.

## 3. Results

The proportion of plasma amino acids is shown in Figure 1. FRE dogs had lower proportions of Asn (*p* = 0.034), His (*p* = 0.009), Gly (*p* = 0.005), Cys-Cys (*p* = 0.028), Leu (*p* = 0.017), and ratio branched-chain amino acids/aromatic amino acids (BCAA/AAA) (*p* = 0.018) when compared to control dogs. However, FRE dogs had a greater proportion of Phe (*p* = 0.013).

Correlations between plasma-free amino acids and plasma fat content, α-tocopherol concentrations, BCS, and CIBDAI indices are presented in Table 2. Total plasma fat content correlated negatively with Trp (r = −0.58, *p* = 0.014) and Phe (r = −0.50, *p* = 0.040). Next, α-Tocopherol (as an indicator of the oxidative status) correlated positively with Asn (r = 0.56, *p* = 0.020), Gly (r = 0.51, *p* = 0.035), Arg (r = 0.65, *p* = 0.004), and Cys-Cys (r = 0.60, *p* = 0.024). On the contrary, Ala correlated negatively with α-tocopherol (r = −0.49, *p* = 0.045). In addition, BCS correlated positively with Leu (r = 0.48; *p* < 0.05). Finally, CIBDAI correlated positively with Phe (r = 0.53, *p* = 0.027), whereas it correlated negatively with Leu (r = −0.69, *p* = 0.002), Lys (r = −0.59, *p* = 0.012), BCAA (r = −0.49, *p* = 0.043), and BCAA/AAA ratio (r = −0.67, *p* = 0.003).

Proportions of faecal amino acids are shown in Figure 2. Faecal Cys-Cys proportion was greater (*p* = 0.005), whereas Phe was lesser (*p* = 0.032) in sick dogs compared to the control. The other faecal amino acids were not statistically affected.

Correlations between faecal amino acids and faecal parameters (fat and humidity), BCS, and CIBDAI indices are shown in Table 3. Faecal fat percentage was negatively correlated with Phe (r = −0.62, *p* = 0.030), Lys (r = −0.65, *p* = 0.022), and Leu (r = −0.57, *p* = 0.050). Faecal humidity percentage correlated positively with Gly (r = 0.59, *p* = 0.045) and negatively with Leu (r = −0.66, *p* = 0.018). No correlation was found with BCS or CIBDAI.

The correlation between faecal amino acids and SCFAs was also investigated (Table 4). The concentration of SCFAs between FRE and healthy dogs was quantified in a previous study [9]. Total SCFAs correlated positively with Phe (r = 0.80, *p* = 0.002), Ile (r = 0.75, *p* = 0.005), and Leu (r = 0.92, *p* = 0.0001); whereas a negative correlation was observed with Gly (r = −0.85, *p* = 0.0005) and Cys-Cys (r = −0.61, *p* = 0.035). The SCFA that presented the highest number of correlations was butyric acid (C4), which correlated positively with amino acids Val (r = 0.67, *p* = 0.017), Met (r = 0.68, *p* = 0.015), Ile (r = 0.77, *p* = 0.003), and Leu (r = 0.80, *p* = 0.001), while negatively with Gly (r = −0.78, *p* = 0.002). Valeric acid (C5) and isovaleric acid (IC5) also presented a high number of correlations. Isovaleric acid correlated negatively with Gly (r = −0.59, *p* = 0.042), Tyr (r = −0.59, *p* = 0.042), and Cys-Cys (r = −0.60, *p* = 0.039), and C5 correlated positively with Leu (r = 0.59, *p* = 0.043) and negatively with Gly (r = −0.61, *p* = 0.034). However, the C2, C3, and IC4 presented a lower number of correlations with faecal amino acids. Thus, C2 correlated negatively with Ser (r = −0.58, *p* = 0.049), and IC4 correlated positively with His (r = 0.64, *p* = 0.026). However, C3 correlated positively with Phe (r = 0.62, *p* = 0.030), and a tendency was observed with Leu (r = 0.56, *p* = 0.060).

Finally, linear adjustments were observed between faecal characteristics (faecal fat or humidity), faecal Leu (R^2^ = 0.32), Phe (R^2^= 0.41), and Gly (R^2^= 0.42) (Figure 3a–c, respectively). These faecal amino acids also presented high linear responses with SCFAs. Thus, more than 70% of the variation of these amino acids in faeces was linearly explained by the concentration of SCFAs: R^2^ = 0.84; R^2^ = 0.70; and R^2^ = 0.70 for Leu, Phe, and Gly, respectively (Figure 3d–f).

## 4. Discussion

Academic literature concerning the amino acid profile in dogs with gastrointestinal disorders is currently quite limited. Some research has been actually carried out in dogs with IRE or PLE [10,11,12,13]. However, current information on the amino acid profile in FRE dogs is not yet available. In the present study, FRE dogs had lower proportions of Asn, His, Gly, Cys-Cys, Leu, and ratio BCAA/AAA in plasma. Studies carried out in human medicine have also shown alterations in the amino acid profile of patients suffering from IBD [5,7], and the benefits of supplementing some amino acids on the reduction of symptomatology have been described [7,18,19]. Also in human studies, some authors have suggested the utility of amino acids like His as monitoring tools for predicting the risk of relapse in patients with ulcerative colitis (UC) [20]. In addition, prior research points to Gly and Cys as important amino acids for the maintenance of oxidative status linked to the inflammatory process, as they are part of antioxidant enzymes such as glutathione [21]. In this study, a positive correlation was found between these amino acids and vitamin E, which is one of the most important antioxidants that participate to a great extent in cell oxidative control in connection with other antioxidant systems to ensure the homeostasis of the individual [16]. Finding lower proportions of Gly and Cys-Cys in FRE dogs could be due in part to the higher use of these amino acids to synthesize glutathione and control the augmented reactive oxygen species (ROS) production that takes place in the inflammatory process. Moreover, these amino acids, together with His, Asn, and Leu, regulate intestinal inflammation, downregulating the production of proinflammatory cytokines [4]. In contrast, Cys, Gly, Asn, and Leu are also responsible for maintaining the normal functioning of the intestinal epithelial barrier by enhancing tight junction proteins [4]. Therefore, the lack of adequate long-term levels would exacerbate the inflammatory process and aggravate the integrity of the mucosal barrier. This lack would thus lead to bacterial adhesion and alteration of transporters responsible for the absorption of nutrients which, in turn, would result in nutritional deficits [22].

It is worth emphasizing that, in the present study, plasma Phe proportion was the only one increased in dogs with FRE. A recent study by Walker et al. [13] also found greater Phe serum concentration in dogs with CEs. It has also been reported that both inflammation and infection often lead to increased levels of Phe in human patients [23] since cytokines induce a strong metabolic disruption, muscle tissue breakdown, and a catabolic state. This state is associated with a higher release and increased Phe plasma levels in demand of the high metabolic rate [24], with Phe being a good indicator of body protein breakdown [25,26]. It is interesting, therefore, to observe that Phe correlated positively with CIBDAI in the present study, indicating a severe state of the disease based on weight and muscle loss. Moreover, the BCAA/AAA ratio decreased in FRE in comparison to that of healthy dogs. Phe is considered an AAA that is converted into Tyr. Branched-chain amino acids (BCAA), including Leu, Ile, and Val, are responsible for regulating the metabolism of glucose, lipid and protein synthesis, intestinal health, and immunity. Thus, BCAAs represent the major nitrogen source for the synthesis of Ala, Gln, and Glu [27] which are essential components for rapidly dividing cells such as enterocytes and immune cells [28]. Other authors reported lower BCAA/AAA in gastrointestinal or hepatic diseases [26] in association with the malnutrition process or with increased protein catabolism [29]. This result was confirmed in the present study by the negative correlation between AAA (Phe) and plasma fat. Some studies carried out in humans found that plasma AAA were higher in obese patients and were positively correlated to adiposity [30,31,32]. In addition, in the present study, Leu, Lys, BCAA, and the BCAA/AAA ratio correlated negatively with CIBDAI. Leu and Lys are essential amino acids, having a significant role in protein anabolism. The amino acid Leu [33] has especially been considered the major regulator of muscle protein synthesis in neonates [34]. Moreover, it has been confirmed that, among the BCAA, the response of muscle protein synthesis is unique to Leu, whereas Val and Ile failed to stimulate protein synthesis activation [35]. This finding is in line with the present results since one of the clinical signs evaluated by the CIBDAI index is weight loss that is associated with muscle loss, a typical sign of CE. Therefore, dogs with greater weight loss would have lower levels of these essential amino acids in plasma and, consequently, a greater CIBDAI classification. According to the results of the present study, Leu was the amino acid that showed the highest correlation, together with BCAA/AAA and the illness state, followed by Lys and Phe. These might then represent potential novel biomarkers for FRE. Commercially available diets containing hydrolysed protein formulated for dogs with CEs do not specify, in most cases, any amino acid profile. More information on the amino acid profile of dogs with FRE could lead to more precise and specific amino acid formulation in dietary interventions, better response to diet, and the recovery of the animal.

Contrary to what was observed in plasma samples, FRE dogs had a lower proportion of Phe in the stool, which might indicate a greater metabolic use of this compound. However, faecal Cys-Cys was high in sick dogs which could be a consequence of an increase of Cys metabolism at this level as reported previously in IBD patients, possibly due to perturbed gut microbiota [36]. The proportions of amino acids were lower in faeces than in plasma as expected, except for the amino acids Asp and Glu. It has been reported that, during acid hydrolysis, the amino acids Asn and Gln are completely converted to Asp and Glu, respectively, while Trp is destroyed [37]. The correlations observed between faecal amino acids and faeces characteristics confirm again the importance of Phe and Leu as possible indicators of intestinal disease severity. Faecal Phe reached the highest negative correlation with the fat proportion in the stool, whereas faecal Leu was negatively correlated with proportions of fat and humidity. Balasubramanian et al. [38] also found lesser levels of Leu, as part of BCAA (Ile, Leu, Val), in the colonic mucosa of IBD patients compared with healthy subjects and considered these as potential biomarkers. Other amino acids involved in the endogenous antioxidant capacity that could be associated with illness, such as Gly, showed a positive correlation with stool humidity in the present study. Bjerrum et al. [39] also found increased levels of Gly in the faecal samples of IBD individuals compared to healthy controls, since this amino acid plays a key role in oxidative homeostasis and the regulation of inflammation [40]. It seems that gut alterations could induce a higher proportion of amino acids involved in oxidative functions such as Cys or Gly, since Cys-Cys was also greater in the stool of FRE dogs. However, no correlations were observed between Cys-Cys and faecal characteristics, although it should be pointed out that both presented a negative correlation with SCFAs.

It has been reported the importance of SCFAs for keeping intestinal health and their levels are reduced in the faeces of adults suffering from IBD [41] or other CEs [9] with numerous studies suggesting that they play an important role in the treatment of inflammation-related diseases [9,41]. Although SCFA production comes mainly from the fermentation of carbohydrates, bacterial fermentation of protein sources serves as well for their obtention [42,43]. It has been reported that protein fermentation by intestinal bacteria in humans could account for 17 % of SCFAs found in the caecum and 38% of SCFAs produced in the rectum [42]. The ratio of available carbohydrates to protein determines substrate utilization by the gut microbiota. Therefore, when energy is scarce, proteins are catabolized by bacteria to produce amino acid-derived end products [43]. In the present research, the amino acid that presented the highest positive correlation with total SCFA proportion was Leu, which was mainly positively related to faecal C4 proportion, followed by Phe, and Ile. Leu and Ile, as BCAA, play an important role in gut health by promoting intestinal development, nutrient transporters, and immune-related function [27]. To the best of our knowledge, there is no previous information on the correlation between faecal SCFAs and faecal amino acids in dogs with CE. According to our results, the higher proportion of faecal Leu and Phe, the higher the faecal SCFAs, which would be associated with better gut health. These results confirm again their importance as indicators of disease severity and faecal characteristics.

## 5. Conclusions

Our results show that dogs with FRE had different plasma and faecal amino acid profiles than control dogs. The high correlation observed between plasma Leu and Phe with CIBDAI suggests that they could be used as disease biomarkers. Furthermore, statistically significant correlations observed for Leu, Phe, Gly, and Cys-Cys with SCFAs might indicate gut microbiota functionality, as well as homeostasis disruption. Consequently, these amino acids might have a role to play in food-responsive enteropathy.

## Figures and Tables

**Figure 1 vetsci-10-00112-f001:**
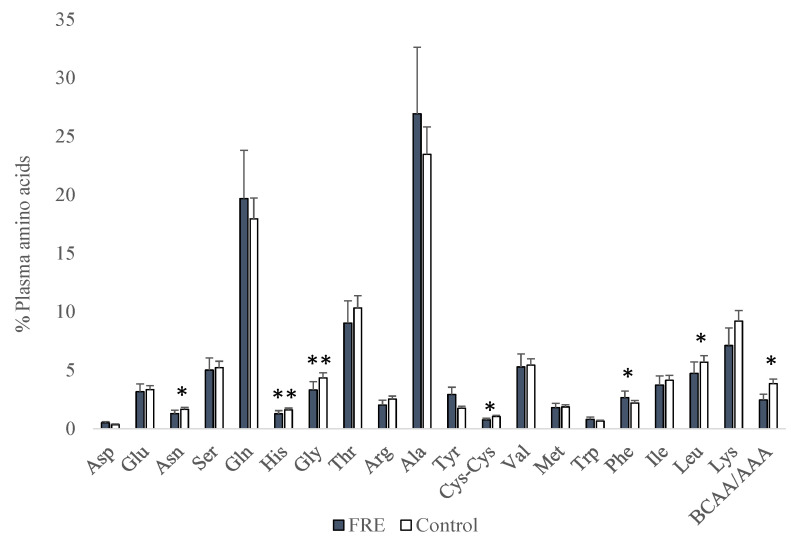
Amino acid proportions (%) in plasma from FRE (food-responsive enteropathy) and control dogs (* *p* < 0.05; ** *p* < 0.01).

**Figure 2 vetsci-10-00112-f002:**
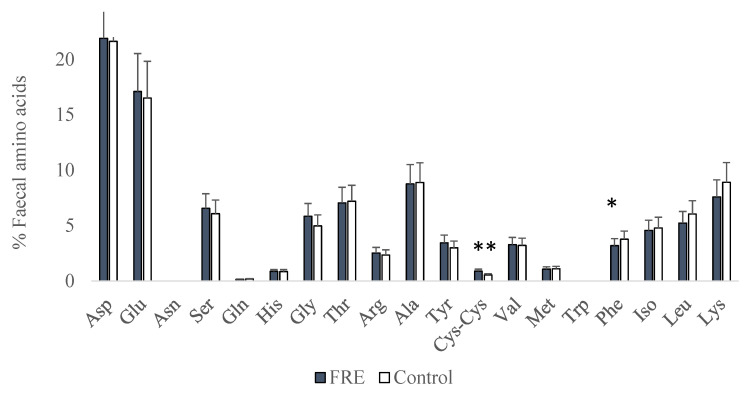
Amino acid proportions (%) in faeces from FRE (food-responsive enteropathy) and control dogs (* *p* < 0.05; ** *p* < 0.01).

**Figure 3 vetsci-10-00112-f003:**
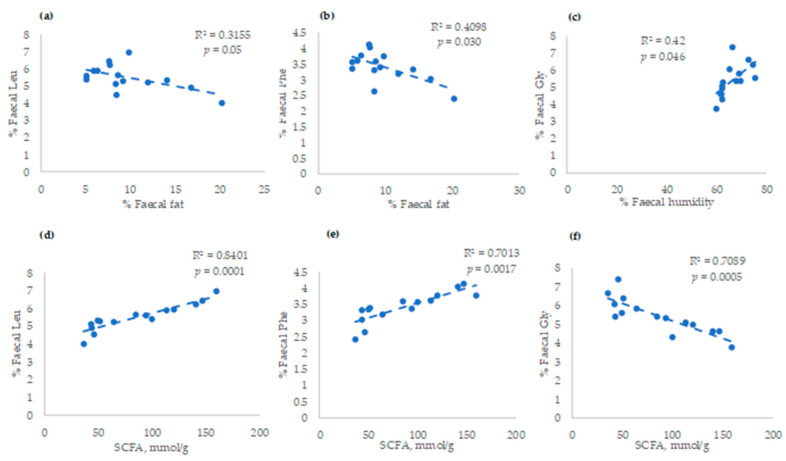
Linear adjustments between (**a**) % fat and Leu; (**b**) % fat and Phe; (**c**) humidity and Gly; (**d**) ∑SCFAs and Leu; (**e**) ∑SCFAs and Phe; and (**f**) ∑SCFAs and Gly in faecal samples.

**Table 1 vetsci-10-00112-t001:** Data of signalment and CIBDAI (canine inflammatory bowel disease activity index) in dogs with FRE (food-responsive enteropathy) and healthy dogs included in the study.

	FRE	Range	Control	Range
	Average ± SD		Average ± SD	
Age (years)	6.2 ± 3.6	(3–13)	4.5 ± 2.1	(3–6)
Male	4 (2 entire, 2 castrated)		3 (3 entire)	
Female	5 (1 entire, 4 spayed)		3 (3 spayed)	
Body weight	11.4 ± 9.4	(2.3–31)	15.5 ± 0.7	(15–16)
Body condition score (scale 9)	4.2 ± 1.4	(2–6)	4.5 ± 0.40	(4–5)
CIBDAI index	7 ± 1.8	(5–10)	0	(0–0)

SD: standard deviation of the mean.

**Table 2 vetsci-10-00112-t002:** Correlation coefficients between plasma amino acids proportions and plasma fat, plasma α-tocopherol, body condition score (BCS), and CIBDAI (canine inflammatory bowel disease activity index).

%	Fat	α-Tocopherol	BCS	CIBDAI
Aspartic acid	−0.39	0.27	0.35	0.19
Glutamic acid	−0.20	0.48	0.28	−0.35
Asparagine	0.04	0.56 ^b^	−0.39	−0.32
Serine	0.41	−0.07	−0.14	−0.07
Glutamine	−0.30	0.26	0.36	0.15
Histidine	0.23	0.15	−0.30	−0.38
Glycine	0.20	0.51 ^b^	0.20	−0.47
Threonine	0.35	−0.02	−0.22	−0.12
Arginine	−0.08	0.65 ^a^	−0.02	−0.42
Alanine	0.15	−0.49 ^b^	−0.22	0.20
Tyrosine	−0.25	0.08	−0.29	0.41
Cystine	−0.13	0.60 ^b^	0.28	−0.40
Valine	−0.13	−0.03	0.28	−0.25
Methionine	0.10	0.18	−0.39	0.08
Tryptophan	−0.58 ^b^	0.06	0.36	0.32
Phenylalanine	−0.50 ^b^	0.03	0.14	0.53 ^b^
Isoleucine	−0.08	0.32	0.12	−0.37
Leucine	0.15	0.38	0.48 ^b^	−0.69 ^a^
Lysine	0.04	0.36	0.43	−0.59 ^b^
BCAA ^1^	0.00	0.26	0.33	−0.49 ^b^
AAA ^2^	−0.32	0.08	−0.24	0.47
BCAA/AAA	0.40	0.03	0.21	−0.67 ^a^

**^1^** BCAA, Branched-chain amino acids: sum of amino acids valine, leucine, and isoleucine. **^2^** AAA, Aromatic amino acids: sum of tyrosine, phenylalanine, and tryptophan. ^a^ Significant at <0.01 probability level (red colour); ^b^ significant at <0.05 probability level (blue colour).

**Table 3 vetsci-10-00112-t003:** Correlation coefficients between faecal amino acids proportions and faeces composition (% humidity, % fat), body condition score (BCS), and CIBDAI (canine inflammatory bowel disease activity index).

% Faeces	% Faecal Fat	% Faecal Humidity	BCS	CIBDAI
Aspartic acid	0.37	0.07	−0.18	−0.12
Glutamic acid	0.25	−0.05	−0.02	0.22
Serine	0.45	0.48	−0.21	0.52
Histidine	−0.13	−0.38	0.33	−0.09
Glycine	0.26	0.59 ^b^	−0.39	0.40
Threonine	0.41	0.29	−0.03	0.15
Arginine	0.29	0.16	0.40	0.23
Alanine	−0.24	0.27	0.13	0.15
Tyrosine	−0.39	0.32	−0.04	0.41
Cystine	0.35	0.35	0.22	0.48
Valine	−0.43	−0.21	0.47	0.13
Methionine	−0.10	−0.34	0.31	−0.1
Phenylalanine	−0.62 ^b^	−0.53	0.28	−0.45
Isoleucine	−0.57	−0.47	0.44	−0.19
Leucine	−0.57 ^b^	−0.66 ^b^	0.44	−0.46
Lysine	−0.65 ^b^	−0.43	−0.02	−0.39

^b^ Significant at <0.05 probability level (blue colour).

**Table 4 vetsci-10-00112-t004:** Correlation coefficients between faecal amino acids proportions and faecal short-chain fatty acids (SCFAs, mmol/g dry matter).

% Faeces	∑SCFAs	C2	C3	IC4	C4	IC5	C5
Aspartic acid	−0.42	−0.12	−0.11	−0.14	−0.54	0.02	−0.24
Glutamic acid	0.04	−0.55	−0.24	0.04	0.37	0.29	0.39
Serine	−0.49	−0.58	−0.53	0.10	−0.03	−0.47	−0.10
Histidine	0.25	−0.33	0.05	0.64 ^b^	0.39	0.13	0.24
Glycine	−0.85 ^a^	−0.10	−0.48	−0.22	−0.78 ^a^	−0.59 ^b^	−0.61 ^b^
Threonine	0.01	−0.24	0.07	−0.19	0.25	0.14	0.01
Arginine	0.27	0.12	−0.04	−0.36	0.42	−0.24	0.32
Alanine	0.14	0.53	0.19	−0.16	0.05	−0.23	−0.26
Tyrosine	−0.15	0.24	−0.29	−0.29	−0.01	−0.59 ^b^	−0.13
Cystine	−0.61 ^b^	−0.42	−0.53	−0.40	−0.23	−0.60 ^b^	−0.16
Valine	0.40	−0.05	−0.01	0.07	0.67 ^b^	−0.24	0.38
Methionine	0.53	−0.24	0.25	0.23	0.68 ^b^	0.34	0.50
Phenylalanine	0.80 ^a^	0.41	0.62 ^b^	0.39	0.56	0.50	0.26
Isoleucine	0.75 ^a^	0.22	0.31	0.06	0.77 ^a^	0.19	0.54
Leucine	0.92 ^a^	0.24	0.56	0.22	0.80 ^a^	0.47	0.59 ^b^
Lysine	0.51	0.48	0.42	0.54	0.12	0.30	0.04

∑SCFAs sum of short-chain fatty acids; DM, dry matter; C2, Acetic acid; C3, propionic acid; IC4, Isobutyric acid; C4, butyric acid; IC5, Isovaleric acid; C5, valeric acid; ^a^ significant at <0.01 probability level (red colour); ^b^ significant at <0.05 probability level (blue colour).

## Data Availability

Data is contained within the article.

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
