# Peer review of "Changes in Faecal and Plasma Amino Acid Profile in Dogs with Food-Responsive Enteropathy as Indicators of Gut Homeostasis Disruption: A Pilot Study"

_vetsci, 2023, doi:10.3390/vetsci10020112_

Round 1
Reviewer 1 Report
The authors suggested amino acid profile alterations as a complementary information for FRE in dogs, with potential to better predict therapeutic outcomes for this disease. The manuscript is well written and the findings are valuable information for FRE diagnostic and therapeutics.
A few comments should be considered:
- more information about how the number of animals was established to obtain statistical significance. The number of animals with FRE is low, considering that the authors used information from several breeds, ages and sexes.
- GLM variable details should also be detailed
- faecal amino acids quantification was performed after owners collection: how was the temperature and other variables (such as the circadian rhythm) controlled for a better reproducibility of these results?
Round 2
Reviewer 2 Report
I still don't understand what is fat percentage (table 2)
Author Response
Dear reviewer,
Please find that "%" (concerning fat percentage) has been deleted in table 2 following your recommedations. This has also been revised in whole paper. Hope that information is presented clearer now. If you have any additional suggestion, please let us to know it.
Many thanks
